# Use of Various Sugarcane Byproducts to Produce Lipid Extracts with Bioactive Properties: Physicochemical and Biological Characterization

**DOI:** 10.3390/biom14020233

**Published:** 2024-02-17

**Authors:** Joana Odila Pereira, Diana Oliveira, Margarida Faustino, Susana S. M. P. Vidigal, Ana Margarida Pereira, Carlos M. H. Ferreira, Ana Sofia Oliveira, Joana Durão, Luís M. Rodríguez-Alcalá, Manuela E. Pintado, Ana Raquel Madureira, Ana P. Carvalho

**Affiliations:** 1CBQF—Centro de Biotecnologia e Química Fina, Laboratório Associado, Escola Superior de Biotecnologia, Universidade Católica Portuguesa, Rua Diogo Botelho 1327, 4169-005 Porto, Portugal; diana.oliveira@pfxbiotech.com (D.O.); afaustino@ucp.pt (M.F.); susanam.vidigal@gmail.com (S.S.M.P.V.); ammpereira@ucp.pt (A.M.P.); chferreira@ucp.pt (C.M.H.F.); assoliveira@ucp.pt (A.S.O.); jdurao@ucp.pt (J.D.); lalcala@ucp.pt (L.M.R.-A.); mpintado@ucp.pt (M.E.P.); rmadureira@ucp.pt (A.R.M.); apcarvalho@ucp.pt (A.P.C.); 2Amyris Bio Products Portugal Unipessoal Lda, 4169-005 Porto, Portugal

**Keywords:** sugarcane, lipid extract, antidiabetic, anticholesterolemic, antihypertension

## Abstract

Sugarcane, a globally cultivated crop constituting nearly 80% of total sugar production, yields residues from harvesting and sugar production known for their renewable bioactive compounds with health-promoting properties. Despite previous studies, the intricate interplay of extracts from diverse sugarcane byproducts and their biological attributes remains underexplored. This study focused on extracting the lipid fraction from a blend of selected sugarcane byproducts (straw, bagasse, and filter cake) using ethanol. The resulting extract underwent comprehensive characterization, including physicochemical analysis (FT-IR, DSC, particle size distribution, and color) and chemical composition assessment (GC-MS). The biological properties were evaluated through antihypertensive (ACE), anticholesterolemic (HMG-CoA reductase), and antidiabetic (alpha-glucosidase and Dipeptidyl Peptidase-IV) assays, alongside in vitro biocompatibility assessments in Caco-2 and Hep G2 cells. The phytochemicals identified, such as β-sitosterol and 1-octacosanol, likely contribute to the extract’s antidiabetic, anticholesterolemic, and antihypertensive potential, given their association with various beneficial bioactivities. The extract exhibited substantial antidiabetic effects, inhibiting α-glucosidase (5–60%) and DPP-IV activity (25–100%), anticholesterolemic potential with HMG-CoA reductase inhibition (11.4–63.2%), and antihypertensive properties through ACE inhibition (24.0–27.3%). These findings lay the groundwork for incorporating these ingredients into the development of food supplements or nutraceuticals, offering potential for preventing and managing metabolic syndrome-associated conditions.

## 1. Introduction

Several pathophysiological disorders are associated with metabolic syndrome, including atherosclerosis, hypertension, obesity, hyperglycemia, hyperlipidemia, and an elevated risk of coronary heart disease and stroke [1,2]. Metabolic syndrome was considered one of the most relevant risk factors for type 2 diabetes and cardiovascular disease in the 21st century [3].

Differently from type 1 (insulin-dependent) diabetes, which occurs when there is deficient insulin production by the pancreas, type 2 (non-insulin-dependent) diabetes results from reduced insulin production, insulin resistance, or ineffective insulin use, representing 90% of people with diabetes. The prevalence of diabetes worldwide in 2019 was 9.3% (463 million people), and this number is expected to increase to 10.9% (700 million) by 2045 [4,5]. Conventional therapies to address this chronic metabolic disease have been reported to be inefficient and with strong adverse side effects, leading to the exploitation of alternative strategies that resort to medicinal plants. Some natural products have been described as cost-effective and with fewer side effects than conventional antidiabetic agents [6,7].

Additionally, patients with diabetes tend to develop hypertension, an associated risk factor in cardiovascular events, contributing to their mortality. High blood pressure further increases the arteriole wall-to-lumen ratio and capillary rarefaction, and endothelium-dependent vascular relaxation is impaired [8,9,10].

Furthermore, despite cholesterol playing a significant role in living cells [11], an elevated level may lead to atherosclerosis [12]. There are three main strategies aiming to reduce bloodstream cholesterol levels: (i) reducing the endogenous production of cholesterol; (ii) reducing the absorption of cholesterol arising both from the diet and endogenous sources discharged from the bile after major meals; and (iii) a combination of these processes. The second strategy involves the reduction in cholesterol absorption through the intestine. One possible strategy is to use cationic resins, such as colestipol, which can sequester bile salts, decreasing their concentration in the intestinal lumen, thus lowering cholesterol solubility, and consequently limiting its absorption through the intestine. Another therapy commonly utilized as a complementary or early preventive strategy is the use of nutraceuticals [13]. Nutraceuticals contain natural compounds generally derived from foods or sources of natural origin intended for prophylactic or therapeutic applications [14].

Sugarcane (*Saccharum officinarum* L.) is a perennial grass used mainly in sugar and ethanol production; during their industrial processing, several byproducts are originated, namely straw (usually left in the field), bagasse (obtained after milling sugarcane), and filter cake (obtained after filtration and purification of the juice of milled sugarcane) (Figure 1).

The global production of sugarcane, a vital commodity in the food and beverage industry and the production of biofuels, has seen significant fluctuations in recent years. In 2021, the Food and Agriculture Organization (FAO) reported that the harvested area of sugarcane was approximately 26 million hectares, with a total production of about 1.859 billion tons per year [15]. However, the international sugar markets are poised for a production deficit in the upcoming 2023/24 (October/September) season. Projections indicate a global output decrease compared to the previous season’s abundant production levels, while world consumption is anticipated to continue its gradual expansion. Consequently, forecasts suggest a decline in sugar inventories during the 2023/24 period. According to the FAO’s preliminary estimate for the 2023/24 season, global sugar production is expected to reach 175.5 million tonnes, reflecting a decrease of 3.6 million tonnes, or 2 percent, from the 2022/23 output. This decrease is primarily attributed to anticipated reductions in outputs from key sugar-producing countries, notably Thailand and India, owing to drier-than-normal weather conditions associated with the prevailing El Niño phenomenon. In Brazil, sugar production for the 2023/24 season is projected to be lower than the record level achieved in 2022/23, while in the United States of America, production is expected to experience a slight decline due to dry weather conditions. In contrast, the European Union is expected to witness a recovery in sugar output for the 2023/24 season compared to the reduced levels observed the previous year, attributed to both increased plantings and higher yields. Similarly, China is anticipated to experience a production rebound driven by robust recovery in sugar crop yields [16]. The demand for sugarcane-based products is increasing worldwide, with Brazil (731 million tonnes) and India (337 million tonnes) representing the principal producers. Thus, the recovery of lipids from these residues, such as policosanols, phytosterols, long-chain aldehydes, and triterpenoids, might create new products for several applications and can be considered a circular economy-based approach [17].

The European Food Safety Authority (EFSA) and the Food and Drug Administration (FDA) recognize phytosterols (e.g., sterols and stanols) to be ingredients that have an impact on the bioavailability of cholesterol, being able to reduce its absorption [13]. Plant sterols and stanols are non-nutritive compounds, structurally analogous to cholesterol, with the same functions in plants as cholesterol in animals, that is, to regulate membrane fluidity and other physiological activities associated with plant biology. Phytosterols inhibit cholesterol absorption, thereby reducing total and low-density lipoprotein (LDL) cholesterol. Although structurally similar to cholesterol, phytosterols are differentiated by their degree of saturation and by their side-chain configuration at the C24 position. There are over 250 phytosterols, the most common being β-sitosterol, campesterol, and stigmasterol [18,19].

Regarding policosanols, they are a mixture of long-chain primary aliphatic alcohols extracted from rice bran, sugarcane, germ, maize, beeswax, apples, grapes, etc., that usually present as a mixture of molecules with carbon chains ranging in length from 24 to 34 carbons known as tetratriacontanol (C34), dotriacontanol (C32), triacontanol (C30), nonacosanol (C29), octacosanol (C28), heptacosanol (C27), hexacosanol (C26), and tetracosanol (C24). Octacosanol constitutes 50–80% of total policosanols [18]. Previous studies revealed that they exhibit several biological functions, including preventing hypercholesterolemia and type 2 diabetes [20], whereas triterpenoids exhibit anti-inflammatory, antioxidant, and anti-diabetic activities [21].

Consumers and companies alike are displaying a growing interest in the procurement of sustainable and secure products. Specifically, there is escalating attention towards products containing innovative “plant-based” constituents. These constituents are sought-after not only for their novelty, but also for their potential to span diverse domains of efficacy, encompassing attributes such as anticholesterolemic, antidiabetic, and antihypertensive functionalities. To date, the available research lacks investigations into the prospective extracts and their corresponding biological attributes. It is worth acknowledging the intricate interplay resulting from the amalgamation and utilization of diverse byproducts.

Thus, the purpose of the present study was to evaluate the physicochemical and biological properties of a lipid extract derived from a mixture of sugarcane byproducts, using advanced analytical techniques and in vitro assays, to target potential nutraceutical applications.

## 2. Materials and Methods

### 2.1. Raw Material—Sugarcane Byproducts

Sugarcane (*Saccharum officinarum* L.) straw, bagasse, and filter cake were provided by Raízen (Guariba, São Paulo, Brazil). The straw and bagasse were dried overnight at 40 °C using a ventilated oven (Memmert GmbH + Co.KG, Schwabach, Germany), followed by a milling process (SM100, Retsch, Vila Nova de Gaia, Portugal) to obtain a particle size < 4 mm. The filter cake was ground with a mortar and dried at 40 °C using a ventilated oven overnight (Memmert GmbH + Co.KG, Schwabach, Germany), and then, stored at room temperature and protected from light until the beginning of the experiment.

### 2.2. Lipid Extraction

The lipophilic extraction was performed using Foss Soxtec TM 8000 apparatus (Hilleroed, Denmark). The dried straw, bagasse, and filter cake samples were mixed at a 1:1:1 ratio and the combined fraction was extracted with ethanol (96% (*v*/*v*), PanReac AppliChem, Barcelona, Spain) at a ratio of 1:20 (*w*/*v*). The extraction temperature was 130 °C, and the extraction period comprised 2 h of boiling and 1 h of rinsing at atmospheric pressure. Afterwards, the solvent was evaporated in a digital rotatory evaporator (Heidolph HeiVAP, Schwabach, Germany) under reduced pressure in a temperature-controlled bath at 40 °C.

To facilitate the handling of the resulting extract, several emulsifiers were tested at different concentrations: carboxymethyl cellulose (Sigma-Aldrich, St. Louis, MO, USA), Tween 80 (Sigma-Aldrich, St. Louis, MO, USA), and lecithin (Alfa Aesar, MA, USA). The one with best performance was Tween 80. The final emulsion was obtained by adding 1% of Tween 80 to the extract, heating it for 15 min at 90 °C, and ultrasonicating it (VCX 130, Sonics & Materials, Newtown, IA, USA) for 2 min at 70% intensity. The emulsion was then used for the subsequent physical and chemical characterization, as well as for biological assays; for the sake of simplicity, from now on, it will be referred to as sugarcane byproduct extract.

### 2.3. Chemical Composition

The composition of the obtained sugarcane byproduct extract was studied with gas chromatography–mass spectrometry (GC–MS model EVOQ; Bruker, Karlsruhe, Germany) analysis following the methodology described by Teixeira et al. [22], which includes a derivatization with N, O-Bis(trimethylsilyl) trifluoroacetamide with 1% trimethylchlorosilane. The equipment used included a GC–QqQ model EVOQ (Bruker, Karlsruhe, Germany) mass spectrometer, with a Rxi-5Sil MS column (30 m × 250 m × 0.25 m nominal). Compound identification was based on a comparison of the obtained mass spectra with the information on the NIST library (v. 2.3).

### 2.4. Physicochemical Characterization

#### 2.4.1. Fourier Transform Infrared Spectroscopy—FT-IR

Sugarcane byproduct extract was analyzed on a PerkinElmer Paragon 1000 FT-IR Spectrometer (Waltham, MA, USA) with the attenuated total reflectance (ATR) accessory. All spectra were acquired with 32 scans and a 4 cm^−1^ resolution, in the region of 4000–550 cm^−1^. All analyses were conducted in duplicate.

#### 2.4.2. Differential Scanning Calorimetry—DSC

The DSC measurements were conducted in a nitrogen atmosphere using DSC 204 F1 Phoenix equipment (Netzsch, Waldkraiburg, Germany), which was calibrated with an indium standard. The samples, ranging from 3 to 6 mg, were placed into aluminum DSC pans and heated from 20 to 500 °C at a rate of 10 °C/min.

#### 2.4.3. Color

Sugarcane byproduct extract color was assessed using a portable CR-410 Chroma meter (from Minolta Chroma, Osaka, Japan) with a *C D65 illuminant, a pulsed xenon lamp light source, an aperture size of 50 mm, a closed cone, and a standard observer of 2° that closely matched CIE 1931 Standard Observer (x̄2λ, ȳλ, z̄λ). The CIELab color scale was employed to measure the degree of lightness (L); redness (+a) or greenness (−a); and yellowness (+b) or blueness (−b) of the samples. Color was measured by placing the sample in a Petri dish on the surface of a white standard plate, with color coordinates of L_standard_ = 93.22, a_standard_ = −0.08, and b_standard_ = 4.04. Each sample was submitted to three readings.

#### 2.4.4. Particle Size Distribution—Mastersizer

Particle size distribution was assessed using a Malvern Mastersizer 3000E for laser diffraction (Malvern Instruments Ltd., Malvern, UK) with the selected parameters including a refractive index of 1.40 and absorption of 0.01. Water was used as dispersant. According to the laser diffraction through the particles of the material, a scattering pattern was generated, and then, used to calculate the particle size via Mie theory. For each sample, three readings were taken.

### 2.5. Biological Characterization—In Vitro Assays

#### 2.5.1. Antidiabetic Potential

##### α-Glucosidase Inhibition Assay

The α-glucosidase inhibition assay was performed according to Kwon et al. [23] with further modifications and optimization. Briefly, 50 μL of the emulsion was mixed with 100 μL of an α-glucosidase solution (1.0 U/mL, Sigma-Aldrich, St. Louis, MO, USA) dissolved in 0.1 M phosphate buffer (pH 6.9). The mixed solution was pre-incubated in a 96-well plate at 25 °C for 10 min. Afterward, 50 μL of p-nitrophenyl-α-d-glucopyranoside (5 mM, Sigma-Aldrich, St. Louis, MO, USA) in 0.1 M phosphate buffer (pH 6.9) was added to each well. The reaction mixtures were incubated at 25 °C for 5 min, and the absorbance readings were recorded at 405 nm by a microplate reader (Synergy H1, Biotek Instruments, Winooski, VT, USA). Acarbose (Sigma-Aldrich, St. Louis, MO, USA) was used as the positive control at a concentration range between 0.08 and 5.30 mg/mL. All assays were performed in triplicate. The α-glucosidase-inhibitory activity was expressed as the α-glucosidase inhibition percentage (%) and calculated as follows:$$\alpha -Glucosidase\;inhibition\;\left(\%\right)=\frac{\left({\Delta Abs}_{control}-{\Delta Abs}_{sample}\right)}{{\Delta Abs}_{control}}\times 100$$
where ΔAbs_control_ is the variation in the absorbance of the control, and ΔAbs_sample_ is the variation in the absorbance of the samples.

##### Dipeptidyl Peptidase-IV (DPP-IV) Inhibition Assay

The DPP-IV assay was performed using a DPP-IV Inhibitor Screening Assay Kit (Cayman Chemical Item No. 700210, Tallinn, Estonia), according to the manufacturer’s instructions. This method uses the fluorogenic substrate Gly-Pro-Aminomethylcoumarin for measuring DPP-IV activity. We mixed 10 μL aliquots of the lipid extract with 10 μL of DPP-IV and 30 μL of assay buffer. The reaction started after the addition of 50 μL of substrate solution (100 mM of Gly-Pro-p-nitroanilide in Tris-HCl buffer (pH = 8.0)) to the microplate and incubated at 37 °C for 30 min; the resulting fluorescence was analyzed using an excitation wavelength of 350–360 nm and an emission wavelength of 450–465 nm (Synergy H1, Biotek Instruments, Winooski, VT, USA). Sitagliptin was the positive control inhibitor (10–100 μM). The analyses were performed in triplicate.

The background fluorescence was subtracted from the 100% initial activity for each inhibitor well. The inhibition rates (%) of the enzymes were calculated as follows:$$DPP-IV\;inhibition\left(\%\right)=\frac{\left(initial\;activity-inhibitor\right)}{initial\;activity}\times 100$$

#### 2.5.2. Anticholesterolemic Potential Assay

The HMG-CoA reductase inhibition assay was conducted using an HMG-CoA Reductase Activity Assay Kit (colorimetric) (Abcam, Cambridge, UK) and performed according to the manufacturer’s instructions. The reaction was monitored for 15 min in a microplate reader at 340 nm (Synergy H1, Biotek Instruments, Winooski, VT, USA), and the absorbance values were plotted against time. The absorbance of two time points within the linear range and with a minimum of 2 min apart were selected and used in the calculations of HMG-CoA reductase activity (units/mg protein) and HMG-CoA reductase inhibition (%), as recommended in the kit. The positive control was a reference drug inhibitor, pravastatin, at a concentration range between 0.60 and 0.19 mg/mL. Analyses were conducted in triplicate in two independent experiments.

#### 2.5.3. Antihypertensive Potential Assay

Angiotensin I-converting enzyme (ACE)-inhibitory activity was evaluated according to Amorim et al. [24], with minor modifications. The substrate of the reaction was o-Abs-Gly-p-nitro-Phe-OH trifluoroacetate salt (Bachem, Bubendorf, Switzerland), and the enzyme was ACE, a peptidyl-dipeptidase A from rabbit lung (Sigma-Aldrich, St Louis, USA) at 42 mU/mL, pH 8.3, with 0.1 mM zinc chloride (Sigma-Aldrich, St Louis, USA). Fluorescence was recorded in kinetic mode using a black polystyrene 96-well microplate (Thermo Fisher Scientific, Waltham, MA, USA) incubated at 37 °C in a microplate reader (Synergy H1, Biotek Instruments, Winooski, VT, USA) for 30 min, using 350 nm for excitation and 420 nm for emission. The control was a commercial inhibitor, captopril (Sigma-Aldrich, St Louis, MO, USA) at a concentration range between 0.09 and 16.29 mg/mL. The percentage of ACE inhibition was calculated using the following formula:$$ACE\;inhibition\;\left(\%\right)=\frac{\left(F_{control}-F_{sample}\right)}{F_{control}}\times 100$$
where F_control_ and F_sample_ are the fluorescence of the control (maximum ACE activity) and sample, respectively. The calculation of IC50 values for the samples was performed through non-linear fitting of the data, using a four-parameter logistic regression model. The assay was performed in duplicate.

#### 2.5.4. Caco-2 Cell Biocompatibility Assay

The biocompatibility evaluation was performed according to the ISO 10993-5 [25] standard, using the PrestoBlue^®^ Cell Viability Reagent (Thermo Fisher Scientific, Waltham, MA, USA), according to the manufacturer’s instructions. The human colon carcinoma (Caco-2; HTB-37^TM^) cells were obtained from the American Type Culture Collection (ATCC) and grown in a humidified atmosphere of 95% air and 5% CO_2_ at 37 °C using high-glucose (4.5 g/L) Dulbecco’s Modified Eagle’s Medium (DMEM) supplemented with 10% (*v*/*v*) heat-inactivated fetal bovine serum (FBS), 1% (*v*/*v*) antibiotic and antimycotic (Gibco, Milan, Italy), and 1% (*v*/*v*) 100× non-essential amino acids (Sigma-Aldrich, St. Louis, MO, USA). Cells were used between passages 36 and 37.

Cells were seeded at 1 × 10^4^ cells/well in a 96-well microplate and incubated overnight at 37 °C. The cells were then exposed to the ST/B/FC extract at different concentrations (i.e., from 0.078 to 2.5 mg/mL) prepared in fresh culture medium, in quadruplicate. Dimethyl sulfoxide (DMSO) (Sigma-Aldrich, St. Louis, MO, USA) at 10% (*v*/*v*) in culture medium was used as a cell death control (negative control) and an unseeded culture medium served as the growth control. After 24 h of exposure, the PrestoBlue^®^ reagent (Invitrogen, Waltham, MA, USA) was added to each well and incubated for 2 h. The fluorescence was recorded (excitation 570 nm; emission 610 nm) after incubation using a microplate reader (Synergy H1, Biotek Instruments, Winooski, VT, USA). The results are expressed as the percentage of metabolic inhibition compared with the control (cells without treatment). At least two independent experiments were performed.

#### 2.5.5. Hep G2 Cell Biocompatibility Assay

The cells of human hepatocellular carcinoma (Hep G2; HB-8065^TM^) were acquired from the American Type Culture Collection (ATCC) and grown in the same culture conditions (37 °C humidified atmosphere with 5% CO_2_). Differently from the culture media used in the previous section, high-glucose DMEM was supplemented with FBS and antibiotic and antimycotic but did not contain the non-essential amino acids. Cells were used between passages 90 and 94.

The biocompatibility assay using Hep G2 cells followed the same protocol as that described previously for Caco-2 cells, based on the ISO 10993-5 [25] standard.

### 2.6. Statistical Analysis

GraphPad Prism 8 software was used to perform a two-way ANOVA with Tukey’s post-test at a 95% confidence level. The results are expressed as mean values ± SD, with significance set at *p* < 0.05. The software also allows for the evaluation of sample normality using the Shapiro–Wilk Test. All the assays were performed in triplicate.

## 3. Results and Discussion

### 3.1. Composition of the Sugarcane Byproduct Extract (GC–MS)

Several authors [26,27] have previously demonstrated the presence of health-beneficial lipophilic compounds in sugarcane byproducts. In the present study, the lipid extract was obtained from a mixture of sugarcane byproducts—straw, bagasse, and filter cake—considering an integrative approach not only in terms of circular economy, but also to enhance their individual potential and develop a unique ingredient with all the properties (i.e., anticholesterolemic, antidiabetic, and hypertensive). The sugarcane byproduct extract was characterized by GC-MS, and the results are shown in Table 1.

Teixeira et al. [22] showed that sugarcane straw and bagasse have the same group of phytochemicals, namely free fatty acids, hydrocarbons, fatty alcohols, sterols, and phenolic compounds. In this study, several chemical classes of alcohols, free fatty acids, fatty alcohols, phytosterols, hydrocarbons, organic acids, fats, sterols, sugars, and fatty aldehydes were detected in a total of 30 compounds identified. The most representative classes were free fatty acids, fatty alcohols, and phytosterols (Table 1).

Within free fatty acids, eight compounds were identified, with the most representative being palmitic acid (11.96 g/kg), octacosanoic acid (6.75 g/kg), and oleic acid (4.65 g/kg). Free fatty acids and their derivatives have a crucial role in several metabolic processes as structural compounds (e.g., in phospholipids), cell energy storage elements (e.g., triglycerides), and signaling compounds in several pathways [28]. Palmitic acid has been reported to significantly inhibit microbial activity against the SARS-CoV-2 virus [29]. In another study, Ledón et al. [30] obtained a mixture of fatty acids from sugarcane wax oil, with the main constituents being palmitic, oleic, linoleic, and linolenic acids. This mixture was evaluated in two models of inflammation, and the results showed that topical application of the mixture presented anti-inflammatory activity without evidence of irritant effects.

Regarding the fatty alcohols group, 1-octacosanol, a straight-chain aliphatic C28 primary fatty alcohol, was the main component, with a concentration of 18.10 g/kg. This compound has been the focus of a considerable amount of research in the last decade due to its potential beneficial effects, such as its capability for reducing cholesterol, antiaggregatory properties, cytoprotective use, and ergogenic properties for human health [31].

Phytosterols were also identified. The principal phytosterols isolated from sugarcane bagasse were campesterol, stigmasterol, and β-sitosterol, as demonstrated previously by Alvarez-Henao et al. [32]. Other authors also found β-sitosterol to be the major phytosterol found in sugarcane leaves and bagasse, followed by stigmasterol [26,33,34]. Phytosterols have been extensively used as a component in food industries due to their health benefits in terms of reducing serum cholesterol levels and the risk of cardiovascular disease [35].

### 3.2. Physicochemical Characterization

#### 3.2.1. FT-IR Analysis

FT-IR analysis enables the recognition of organic functional groups demonstrating correspondence to their respective composition [22]. The vibrational bands of the sugarcane byproduct extract described in Figure 2 were identified based on [36].

The vibrational band at 3306 cm^−1^ is related to the -OH stretching and bending vibrations and C-O asymmetric and symmetric stretching vibrations, respectively. These vibrations can be produced by alcohol groups, which agrees with GC–MS results that identified several fatty alcohols and phytosterols. Moreover, the FT-IR spectra also showed one vibrational band related to amine groups at 1635 cm^−1^, corresponding to the RONH_2_ functional group.

#### 3.2.2. Thermal Analysis

The thermal properties of the sugarcane byproduct extract were studied using differential scanning calorimetry (DSC), as depicted in Figure 3.

The DSC results showed a peak at 100 °C. In the study of Goh et al. [37], it was found that among the four major transitions, the transition at 100 °C was associated with protein denaturation. This indicates that the peak at 100 °C in the DSC results could be attributed to the denaturation of proteins, rather than a direct characteristic of the lipid composition.

#### 3.2.3. Color

Color is a crucial parameter for the packaging of light-sensitive materials and from a consumer’s standpoint [38]. The sugarcane byproduct extract showed high brightness values (L* 82.97 ± 0.71), demonstrating that they are clear and transparent. The values of a* and b* are −4.58 ± 0.25 and 22.43 ± 0.84, respectively, indicating that the color is closer to green and yellow, which can also be validated with naked eye, as shown in Figure 4.

#### 3.2.4. Particle Size Distribution

The particle size distribution of the sugarcane byproduct extract was assessed by Mastersizer. The Dx 10 (µm), Dx 50 (µm), Dx 90 (µm), and Span were 0.315 ± 0.007, 0.582 ± 0.065, 1.491 ± 0.165, and 2.123 ± 0.395, respectively. Briefly, most of the particle size distribution is under 1.5 µm. These results suggest small particles (around 0.5 µm) and a monodisperse sample. Gu et al. [39] demonstrated a droplet size distribution of sugarcane wax-based emulsion gels under 5 μm, with an average of 2.31 μm, larger than our extract. On the other hand, Ishaka et al. [40] showed that policosanol emulsion has a nano-size particle distribution below 0.1 μm, smaller than ours. These values are consistent with global emulsions, which contain dispersed droplets with a mean diameter usually in the range of 0.1–100 μm [41].

### 3.3. Biological Characterization

#### 3.3.1. Antidiabetic Potential

In this study, the antidiabetic potential of the sugarcane byproduct extract was assessed using two different assays: the α-glucosidase inhibition and the DPP-IV inhibition assay. α-Glucosidase is an enzyme located in the epithelium of the small intestine, responsible for the hydrolysis of oligosaccharides and disaccharides into monosaccharides [42]. The inhibition of α-glucosidase activity theoretically delays carbohydrate absorption, managing hyperglycemia and, thereby, helping to manage type 2 diabetes. Several plant-derived polyphenols have been shown to have inhibitory effects on α-glucosidase, reducing the digestion rate of complex starches and oligo, tri-, and disaccharides into absorbable glucose [43]. The sugarcane byproduct extract was assessed for α-glucosidase inhibition and compared with acarbose, used as a positive control. The results showed that the extract presented an α-glucosidase-inhibitory effect between 5 and 60%, in a concentration range between 1.88 and 15.0 mg/mL (Figure 5). Acarbose presented α-glucosidase inhibition of 60%, comparable to the highest concentrations (i.e., 7.50–15.0 mg/mL) assessed. Although acarbose is commercially available and commonly administrated to treat type 2 diabetes, several side effects, such as abdominal discomfort, flatulence, and diarrhea, have been associated with its use [44].

DPP-IV is a glycoprotein present on the surface of most cell types and is associated with immune regulation, signal transduction, and apoptosis. In diabetic patients, glucose tolerance improved upon increasing the activity of glucose-regulating hormones, namely glucagon-like peptide-1 (GLP-1) and glucose-dependent insulinotropic polypeptide (GIP) [45]. DPP-IV is the enzyme responsible for the degradation of GLP-1 and GIP [46]. Gliptins, such as sitagliptin, vildagliptin, saxagliptin, alogliptin, linagliptin, gemigliptin, and teneligliptin, are commercialized in the USA, Europe, Japan, and Korea as DPP-IV inhibitors, although they also present side effects [47]. Natural alternatives to extend the activity of GLP-1 and GIP without side effects are under investigation.

The extract tested in concentrations between 1.88 and 15.0 mg/mL (Figure 6) exhibited a strong capacity to inhibit DPP-IV enzymes (i.e., 25–100%). Moreover, sitagliptin, used as a positive control, inhibited DPP-IV activity by 100%. Similar findings were obtained by Oliveira et al. [48] using a polyphenolic extract derived from sugarcane straw that exhibited a strong capacity to inhibit the DPP-IV enzyme (i.e., 62–114%).

#### 3.3.2. Anticholesterolemic Potential

The mevalonate pathway, responsible for cholesterol production, depends on the enzyme 3-hydroxy-3-methyl-glutaryl-coenzyme A (HMG-CoA) reductase. Cholesterol synthesis in the liver is reduced by HMG-CoA reductase inhibition [49]. The cholesterol-lowering effect of policosanols present in the extract from sugarcane was previously reported in animal models. In rats, policosanol orally administered (500 mg) for four weeks reduced cholesterol synthesis by affecting HMG-CoA reductase [50]. Furthermore, in humans with normal or borderline total cholesterol levels, policosanol significantly decreased low-density lipoprotein cholesterol (LDL-c), total cholesterol, and triglyceride levels, and increased high-density lipoprotein cholesterol (HDL-c) [51,52,53].

Among the five concentrations tested, only the three highest showed more than 50% inhibition of HMG-CoA reductase activity (Figure 7). However, since the concentration of 2.5 mg/mL was revealed to be cytotoxic (as shown in 3.8), only concentrations of 0.63 and 1.25 mg/mL are adequate to be used safely. The extract at 1.25 mg/mL showed an inhibitory effect of 59.2%, and pravastatin, used as positive control, showed enzyme inhibition of 75.8%. Therefore, policosanols could be a sustainable alternative to simvastatin or pravastatin, used today as HMG-CoA reductase inhibitors [54].

#### 3.3.3. Antihypertensive Potential

ACE is a hydrolase with a vital role in regulating blood pressure, and treatments for hypertension include blocking this specific enzyme [55]. The potential to inhibit hypertension-related ACE was screened in the sugarcane byproduct extract, and all the concentrations tested (i.e., 0.08–1.25 mg/mL) exhibited moderate inhibition of ACE, with the concentration 1.25 mg/mL (again, the highest non-cytotoxic value) showing the highest inhibition (i.e., 27.2%) (Figure 8). To the best of our knowledge, this study was the first to attempt assessing the antihypertensive potential of sugarcane byproducts.

### 3.4. Cytotoxicity

#### In Vitro Assay in Caco-2 and Hep G2 Cells

Biocompatibility in Caco-2 cells was assessed by determining the safe and sublethal concentrations of the extract. Caco-2 cells were used due to their relevance in an analysis of the impact of food components in the intestinal epithelium [56]. The results showed that the extract was not cytotoxic under 1.25 mg/mL for CaCo-2 cells (Figure 9A); however, at 2.5 mg/mL, toxicity was detected after 24 h of exposure.

Similarly, the biocompatibility assay in Hep G2 cells aimed to detect the safe and sublethal concentration of the sugarcane byproduct extract. Since the build-up of lipids in hepatocytes is a pathologic characteristic, it is crucial to investigate how the extract affects these cell types [57]. Again, a concentration under 1.25 mg/mL was considered non-cytotoxic after the 24 h treatment (Figure 9B).

## 4. Conclusions

This work provides additional information for the future use of more sustainable and natural sources of bioactive compounds to replace those currently in use for metabolic syndrome prevention, such as synthetic drugs, which have been shown to exert harmful secondary effects. In addition, the contribution to a circular economy framework, by recovering underused but valuable byproducts, should also strengthen this line of research. The produced sugarcane byproduct extract has shown interesting potential as a multifunctional phytochemical ingredient in the prevention and amelioration of several metabolic disfunctions. The presence of health-promoting compounds such as phytosterols and 1-octacosanol is most likely responsible for the antidiabetic, anticholesterolemic, and moderate antihypertension properties of this extract. Nevertheless, further testing should be carried out to guarantee its safe application in humans, namely assessing these biological properties after the extract’s digestion and absorption process, i.e., their bioavailability.

## Figures and Tables

**Figure 1 biomolecules-14-00233-f001:**
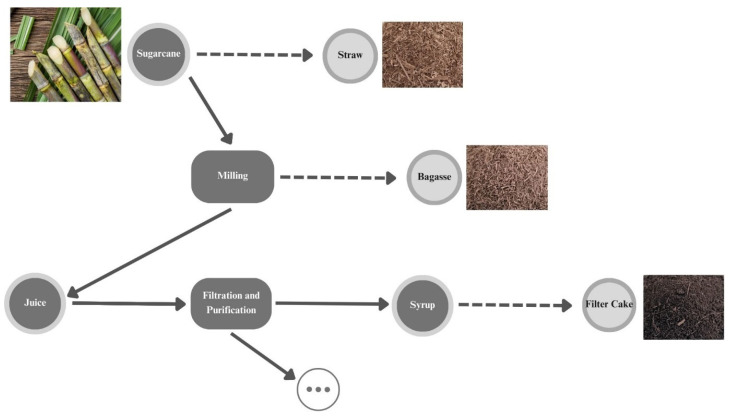
Byproducts (dash arrows) of sugarcane process.

**Figure 2 biomolecules-14-00233-f002:**
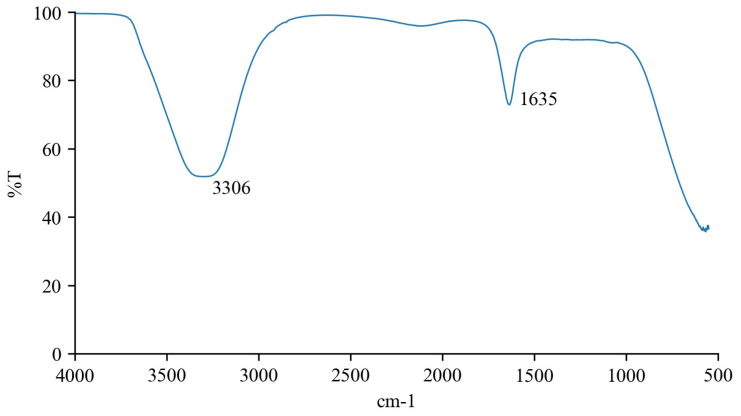
FT-IR emulsion spectra of sugarcane byproduct extract.

**Figure 3 biomolecules-14-00233-f003:**
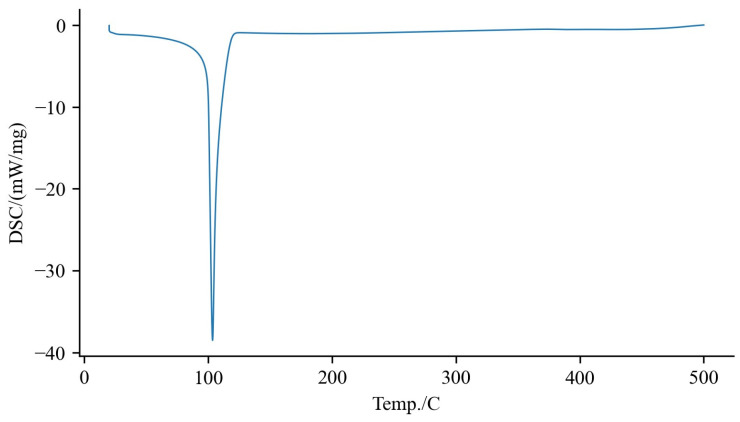
DSC thermograms of sugarcane byproduct extract obtained at a heating rate of 10 °C/min under a nitrogen atmosphere.

**Figure 4 biomolecules-14-00233-f004:**
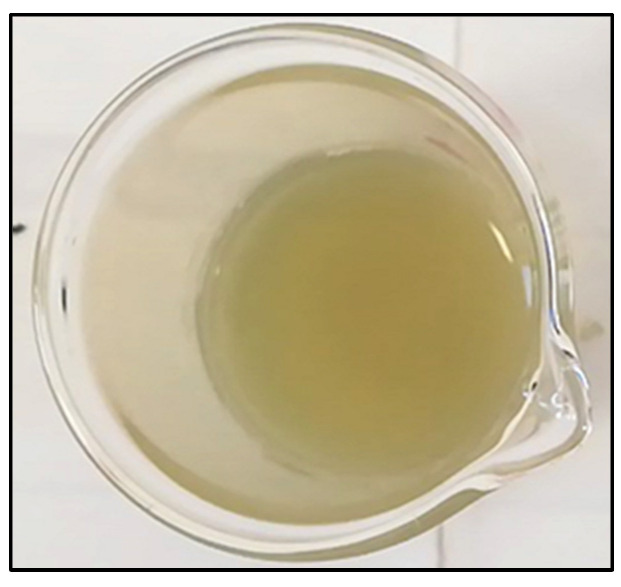
Sugarcane byproduct extract homogenized with 1% Tween 80.

**Figure 5 biomolecules-14-00233-f005:**
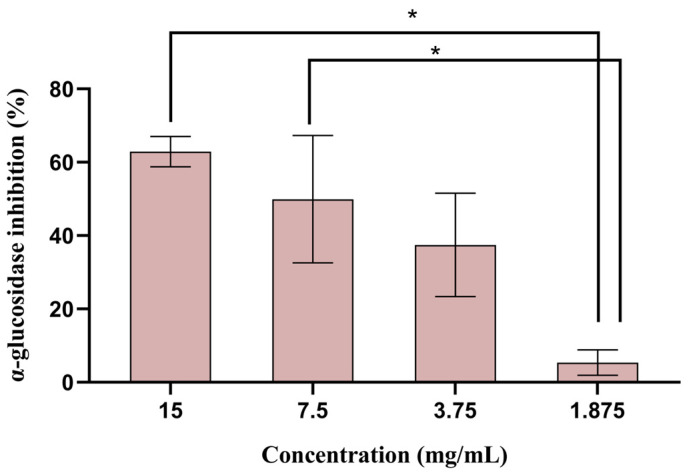
Percentage of α-glucosidase inhibition (mean ± SD) tested with different sugarcane byproduct extract concentrations. Acarbose (1.33 mg/mL) was used as a positive control. * *p* < 0.05 indicates statistically significant differences observed for the different concentrations.

**Figure 6 biomolecules-14-00233-f006:**
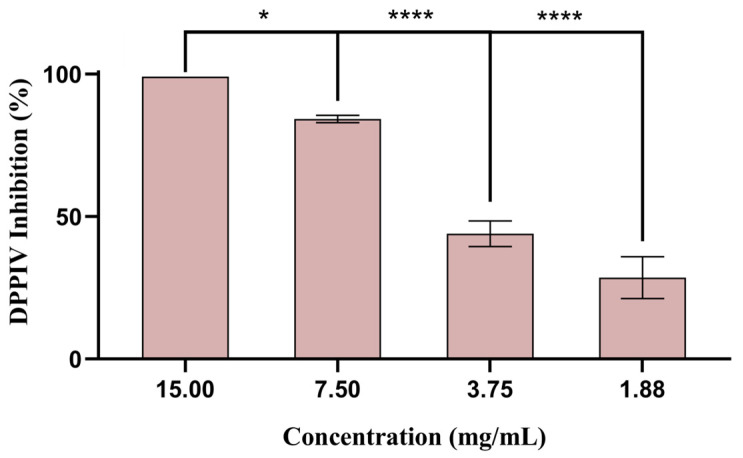
Percentage of DPP-IV inhibition (mean ± SD) tested with different sugarcane byproduct extract concentrations. Sitagliptin (0.04 mg/mL) was used as a positive control. * *p* < 0.05 and **** *p* < 0.0001 indicate statistically significant differences observed for the different concentrations.

**Figure 7 biomolecules-14-00233-f007:**
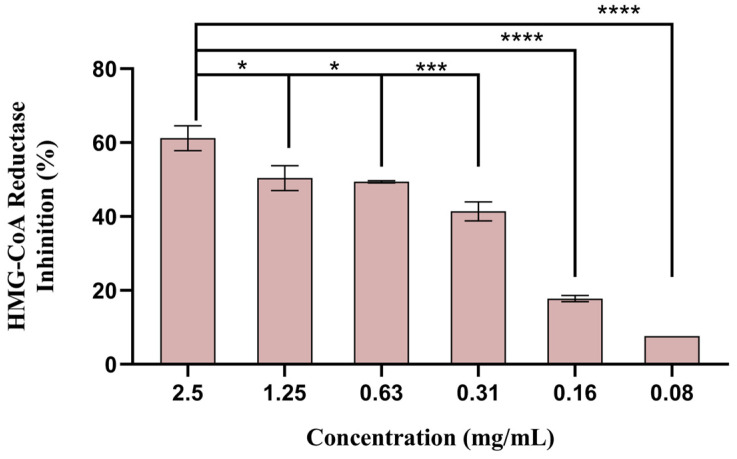
Percentage of HMG-CoA reductase inhibition (mean ± SD) tested with different sugarcane byproduct extract concentrations. Pravastatin (0.60 mg/mL) was used as a positive control. * *p* < 0.05, *** *p* < 0.001, and **** *p* < 0.0001 indicate statistically significant differences observed for the different concentrations.

**Figure 8 biomolecules-14-00233-f008:**
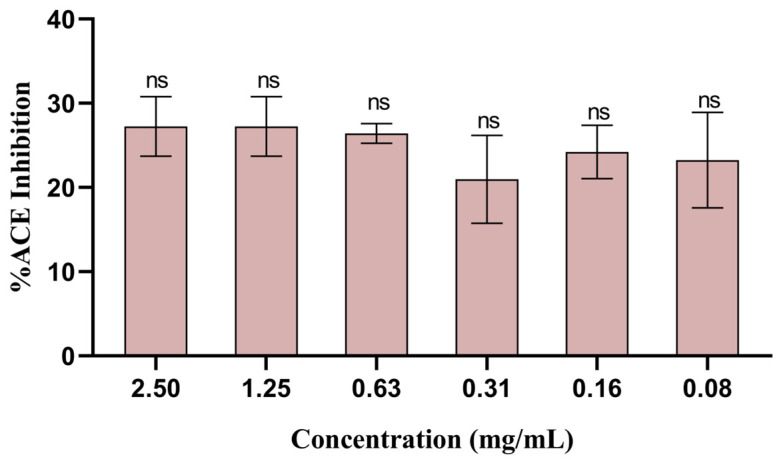
Percentage of ACE inhibition (mean ± SD) tested with the sugarcane byproduct extract at different concentrations. Captopril (2.17 mg/mL) was used as a positive control. ns indicates statistically non-significant differences observed for the different concentrations.

**Figure 9 biomolecules-14-00233-f009:**
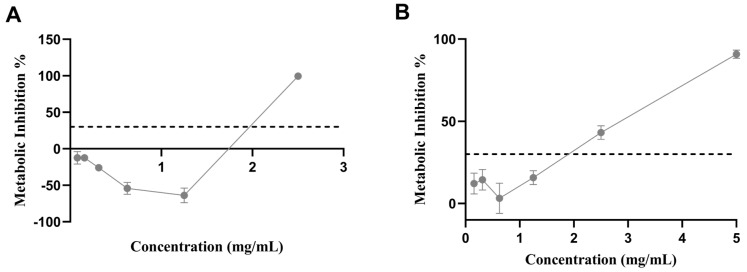
Metabolic inhibition profile of (**A**) Caco-2 cells and (**B**) Hep G2 when exposed to sugarcane byproduct extract. The dotted line represents the 30% cytotoxicity limit as defined by the ISO 10993-5 [25]. Values expressed are the mean ± standard deviation.

**Table 1 biomolecules-14-00233-t001:** Composition profile (g/kg) of alcohols, free fatty acids, fatty alcohols, phytosterols, hydrocarbons, organic acids, fats, sterols, sugars, and fatty aldehydes of sugarcane byproduct extract (mix of ST/B/FC). Values are presented as mean ± standard deviation.

Sugarcane Byproduct Extract (g/kg)	
2,3-Butanediol	0.29 ± 0.04
Acetin	1.23 ± 0.08
Glycerol	4.60 ± 0.57
1,2,3-Butanetriol	0.36 ± 0.03
Ribitol	0.17 ± 0.05
ΣAlcohol	6.66 ± 0.62
Palmitic acid	11.96 ± 0.39
Linoleic acid	1.86 ± 0.15
Oleic acid	4.65 ± 0.31
Stearic acid	3.63 ± 0.41
Arachidic acid	0.62 ± 0.04
Lignoceric acid	0.97 ± 0.12
Octacosanoic acid	6.75 ± 1.09
Triacontanoic acid	1.79 ± 0.39
ΣFree fatty acid	32.23 ± 2.05
1-Hexacosanol	2.22 ± 0.17
1-Octacosanol	18.10 ± 2.19
1-Triacontanol	4.07 ± 0.46
1-Dotriacontanol	2.86 ± 0.35
ΣFatty alcohol	27.25 ± 3.14
Heptacosane	1.80 ± 0.10
Nonacosane	0.65 ± 0.04
ΣHydrocarbon	2.45 ± 0.13
Lactic acid	3.35 ± 0.07
β-Hydroxypyruvic acid	0.31 ± 0.05
ΣOrganic acid	3.66 ± 0.11
Campesterol	4.06 ± 0.37
Stigmasterol	6.25 ± 0.56
β-Sitosterol	9.45 ± 0.67
Stigmast-4-en-3-one	0.81 ± 0.10
ΣPhytosterols	20.56 ± 1.69
ΣSugars	8.86 ± 2.41
Octacosanal	3.38 ± 0.34
(E)-9-Octadecenoic acid ethyl ester	0.71 ± 0.06
4-Coumaric acid	2.83 ± 0.26

## Data Availability

Data are unavailable due to privacy restrictions.

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
