# Peer review of "Use of Various Sugarcane Byproducts to Produce Lipid Extracts with Bioactive Properties: Physicochemical and Biological Characterization"

_biomolecules, 2024, doi:10.3390/biom14020233_

Round 1
Reviewer 1 Report
Comments and Suggestions for Authors
This article is very interesting, but I have a few questions. Can you answer the following?
1. The solvent was extracted with 96% ethanol. Why Lipid extraction?
2. Why are straw, bagasse, and filter cake indicated in table1? The title says sugarcane extract.
3. The compound presented in table 1 is the main extract material. Is there an efficiency in obtaining the material if it is to be industrialized later?
4. Particle size appears to have been added to the test items based on cell permeability among physical properties.
Is the particle size meaningful in the anti-diabetic, anti-hypertensive, etc. presented in this experiment?
5. Can each experiment presented in this experiment represent the disease? (Example) antidiabetic- α-glucosidase inhibition, Antihypertensive - ACE inhibition)
Comments on the Quality of English LanguageThere is no part of what you wrote that I don't understand.
No problems were identified in the English sentences.
Reviewer 2 Report
Comments and Suggestions for Authors
In the paper entitled “Use of various sugarcane byproducts to produce lipid extracts with bioactive properties: physicochemical and biological characterization”, the lipid fraction was extracted from a blend of selected sugarcane byproducts (straw, bagasse, and filter-cake) using ethanol, and its chemical composition was assessed by GC-MS. In addition, the physicochemical properties and bioactivities of the resulting extract were also evaluated. It was found that the extract exhibited certain antidiabetic effects by inhibiting the activities of α-glucosidase and DPP-IV, anticholesterolemic potential with HMG-CoA reductase inhibition, and antihypertensive properties through ACE inhibition. This work could provide a theoretical basis for the development of food supplements and nutraceuticals, so I recommend that this manuscript be accepted for publication in your journal after a major revision.
1) The extract containing diverse compounds displayed various biological activities. What were the action mechanisms of the extract and its interactions with the potential bio-targets?
2) Please check carefully the figure in the manuscript. There was no Figure 2.
3) Please check carefully the results presented in Figure 1, because too few characteristic signals occur in the FT-IR spectrum of the extract containing diverse compounds.
4) The English writing of the manuscript should be improved.
5) All of the bioassays for the extract were conducted in vitro. The authors should supplement some in vivo bio-tests.
6) More physicochemical properties of the extract should be assessed.
Comments on the Quality of English LanguageThe English writing of the manuscript should be improved.
Reviewer 3 Report
Comments and Suggestions for Authors
Reviewing the manuscript entitled "Use of various sugarcane byproducts to produce lipid extracts with bioactive properties: physicochemical and biological characterization," the authors in this study explore the biological properties of extracts from sugarcane byproducts, including straw, bagasse, and filter cake. The lipid fraction was extracted using ethanol and characterized for antihypertensive, anticholesterolemic, and antidiabetic properties. The extract showed significant antidiabetic effects, inhibiting α-glucosidase and DPP-IV activity. The findings could be used in food supplements or nutraceuticals to prevent manage metabolic syndrome-associated conditions.
Although the present study is quite interesting because it introduces the concept of using by-product compounds. I am quite wary of publishing this work because it presents some serious problems:
· The bibliography should be entered in the format required by the journal.
· Plant species names should be in italics.
· More recent data from the FOA should also be entered in the introduction.
· The compounds mentioned in the introduction should be presented with a figure.
· Needs better image and graphics quality.
· There are errors in the numbering of the images.
· In the diagrams there should be a marking of the significance of the results.
· Figure 5 and 6 can be merged into one image and a comparison of the results for the two cell lines can be performed.
· The discussion of results should be a separate section from the results.
· Extensive editing of the English language is required.
For the above reasons, I propose to address the above major issues and review the manuscript again.
Comments on the Quality of English LanguageExtensive editing of the English language is required.
Round 2
Reviewer 2 Report
Comments and Suggestions for Authors
The authors have made substantial revision for this manuscript, so I recommend that it could be accepted for publication in this journal.
Comments on the Quality of English LanguageThe English writing can be improved.
Reviewer 3 Report
Comments and Suggestions for Authors
The revised version of the manuscript is in complete agreement with me. The authors answered all the questions I put to them. I recommend publishing the article.